# Progress and Perspectives in Colon Cancer Pathology, Diagnosis, and Treatments

**DOI:** 10.3390/diseases11040148

**Published:** 2023-10-24

**Authors:** Noor Alrushaid, Firdos Alam Khan, Ebtesam Al-Suhaimi, Abdelhamid Elaissari

**Affiliations:** 1Universite Claude Bernard Lyon-1, CNRS, ISA-UMR 5280, 69100 Villeurbanne, France; naalrushaid@iau.edu.sa; 2Department of Stem Cell Research, Institute for Research and Medical Consultations (IRMC), Imam Abdulrahman Bin Faisal University, Dammam 31441, Saudi Arabia; fakhan@iau.edu.sa; 3Biology Department, College of Science, Institute of Research & Medical Consultations (IRMC), Imam Abdulrahman Bin Faisal University, Dammam 31441, Saudi Arabia; ealsuhaimi@iau.edu.sa

**Keywords:** colorectal cancer, development of colorectal cancer, colorectal cancer pathology, colorectal cancer diagnosis, molecular markers, cancer treatments

## Abstract

Worldwide, colon cancer is the third most frequent malignancy and the second most common cause of death. Although it can strike anybody at any age, colon cancer mostly affects the elderly. Small, non-cancerous cell clusters inside the colon, commonly known as polyps, are typically where colon cancer growth starts. But over time, if left untreated, these benign polyps may develop into malignant tissues and develop into colon cancer. For the diagnosis of colon cancer, with routine inspection of the colon region for polyps, several techniques, including colonoscopy and cancer scanning, are used. In the case identifying the polyps in the colon area, efforts are being taken to surgically remove the polyps as quickly as possible before they become malignant. If the polyps become malignant, then colon cancer treatment strategies, such as surgery, chemotherapy, targeted therapy, and immunotherapy, are applied to the patients. Despite the recent improvements in diagnosis and prognosis, the treatment of colorectal cancer (CRC) remains a challenging task. The objective of this review was to discuss how CRC is initiated, and its various developmental stages, pathophysiology, and risk factors, and also to explore the current state of colorectal cancer diagnosis and treatment, as well as recent advancements in the field, such as new screening methods and targeted therapies. We examined the limitations of current methods and discussed the ongoing need for research and development in this area. While this topic may be serious and complex, we hope to engage and inform our audience on this important issue.

## 1. Introduction

Among different types of cancer, colorectal cancer (CRC) is classified as the third most commonly reported cancer after breast and lung cancers and is ranked as the second most common cause of cancer death [1]. CRC is more frequently diagnosed in males than in females [2]. It could be in the colon, anus, or rectum; all these distinct cancer types are indicated as either colon or colorectal cancer. As per the International Agency for Research on Cancer (IARC), it has been estimated that between 2020 and 2040 the global liability of colorectal cancer will grow by 56%, meaning more than 3 million cases per year. The estimated number of death cases from this cancer will rise to 69% and about 1.6 million death cases in 2040. This increase will be mostly in populations with a high human development index [3]. Generally, colon cancer develops when healthy colon cells’ DNA undergoes alterations (mutations). The DNA of a cell carries a collection of instructions that tell the cell what to perform. It has four stages; in stages I and II, healthy cells spread and divide regularly to keep bodily processes on the colon wall functioning appropriately. Stage III cancer cells continue to proliferate and divide even when more cells are not required because of DNA damage. These cells multiply to create a tumor, and it spreads to nearby lymph nodes. Cancer cells can grow to attack and damage the normal tissue nearby, as shown in Figure 1. They then spread to different parts of the body (metastasis) in stages III and IV [4]. This kind of cancer is annually increasing worldwide in terms of its incidence rates, especially in Western countries, and has many risk factors. There are several complex factors to occur, like genetic/epigenetic, environmental, and lifestyle factors. To illustrate, according to the American Cancer Society, the family history, and genes coding for certain diseases, like diabetes and being overweight, are the risk factors for cancer, and people who are not undertaking physical activity and regularly eating red meat may have higher chances of cancer development. Moreover, working during the night, smoking and overdrinking alcohol also may cause cancer death [5].

CRC can be easily treatable if detected in its early stages; however, some of the cases need 15 years to develop, while others are faster [1]. Early diagnosis at stages 0, I, or II increases the chance of survival rate to more than 80%; however, in the late-stage cases (III or IV) when cancer has metastasized, the chance of survival is dropped to less than 10% [6]. The diagnostic methods to detect colon cancer are colonoscopy, stool tests, occult blood tests, and fecal immunochemical tests. Many studies work to reduce the use of colonoscopy, as it is expensive and may cause some health risks, such as perforation of the colon, post-polypectomy, intra-peritoneal bleeding, and other infections [1]. This review highlighted the most used diagnostic methods for colon cancer and how we can detect this cancer in an early stage. Moreover, there are many types of colon cancer; the common one is adenocarcinoma, which makes mucus in the colon or rectum. The less common types are lymphomas in lymph nodes, carcinoids that arise from hormone-making, sarcomas in soft tissues, and gastrointestinal stromal tumors in the digestive tract [7]. This review covered many areas, including the diagnosis and differential treatment for colon cancer. The ratio of new cancer cases for the year 2020 for colon cancer is shown in Figure 2 [5]. In this review, we emphasized several aspects of colon cancer development, diagnosis, and cancer therapies.

## 2. Risk Factors for CRC Development

CRC is considered the second most common cancer for males and females, and some studies found that males were significantly more impacted than females. The risk of CRC increases substantially after the age of 50 years, and the ratio increases at the age of 40 years [8]. There are many risk factors responsible for the development of this disease. The most important factors are the family history and the medical history of the patient (Table 1). The genetic factor increases the risk of developing colon cancer, and it includes the number of relatives and relative relations [9]. Moreover, a medical history of having inflammatory bowel disease (IBD), ulcerative colitis, or Crohn’s disease increases the danger of colorectal cancer. Also, patients who have had previous cancer treatments or have diabetes are more likely to get colon cancer and need to undergo screening constantly [5]. In some Japanese and Korean studies, it was found that body height is one of the risk factors for CRC in both men and women [10]. Some studies also found that being overweight and obese raised the risk of colon cancer. There is also an association between diet and lifestyle with CRC. The consumption of high amounts of red meat also causes this disease, but regular exercise and physical activity may delay CRC [11]. Additionally, drinking alcohol and smoking increases the severity of injuries and cancer-led deaths. As per the International Agency for Research on Cancer, the World Health Organization (WHO), as shown in Figure 3 the cases of CRC have dramatically increased in the far northern and southern areas of the world compared with the middle area. Africa and India’s regions have the lowest ratio of cases [12]. The incidence of colon and rectal cancer increases with age, but about 40% of cases are diagnosed after the age of 75 [13,14]. About 40% of patients are diagnosed with the disease at stages I-II, 40% at stage III, and 20% at stage IV [15,16]. Patients with a history of colonic polyps and patients with a history of colon or rectal cancer or colonic polyps in close relatives are at an increased risk [17,18]. Inflammatory diseases of the intestine, i.e., ulcerative colitis and Crohn’s disease, increase the risk of colon cancer [19]. It is believed that the probability of colon cancer in such patients is around 15–20% after 30 years, and the risk is possibly higher with ulcerative colitis [19]. Tobacco smoking appears to increase both the risk of colon and rectal cancer, as well as the death rate caused by it [20]. Physical activity [21] daily aspirin use [22], and vitamin D with calcium appear to have a protective effect [23]. An increased consumption of vegetables and fruits, along with reduced meat consumption, seems to potentially reduce the risk of cancer, especially in the left part of the colon [13].

## 3. Developmental Stages of CRC

The stages of colon cancer are classified into three stages, sometimes collectively named a TNM system, which was prepared by the Union for International Cancer Control and later accepted by the American Joint Committee on Cancer [5,12]. An example of a primary tumor (T), which indicates the tumor size and area of growth, along with the number of layers attacked by the tumor, is shown in Figure 4.

The regional lymph nodes (N), where the tumor spreads to nearby lymph nodes, are shown in Figure 5. The distant metastases (M), where the tumor spreads to other body organs like the lungs or the liver, are shown in Figure 5. Normally, the category has a number with the letter, for example, M2, to provide more information about the cancer stage [24]. This number indicates how far the cancer is in the walls or organs, which is as follows: Stage 0 indicates the earliest stage and means no growth in the mucosa or innermost layer of the colon. In stage 1, the tumor reaches the submucosa, but nothing spreads to the lymph nodes. The next stage is stage 2; in this stage, the tumor develops but still does not expand to the lymph nodes, this stage has three sub-groups, namely stage 2A, when it moves to the outer layers of the colon, but the growth does not completely, stage 2B, which indicates that the tumor has reached the visceral peritoneum, and stage 2C, where the cancer cell has grown on nearby organs or structures. Stage 3 also has three categories. In stage 3A, the tumor spreads to nearby lymph nodes, and stage 3B has two classifications depending on how many lymph nodes are reached, i.e., three, four, or more. Stage 3C denotes when the tumor has grown outside the muscular layers and is found in four or more adjacent lymph nodes but in the implicated area. The last one is stage 4, which also has three different stages. In stage 4A, the tumor is moved to one distant site, such as the liver, lungs, or lymph nodes. Stage 4B is when it reaches two or more distant sites; the last category, stage 4C, indicates that the cancer has been scattered across the peritoneum [24]. Moreover, CRC is classified according to the cancer cell compared with the normal cell under the microscope and is graded from 1–4; low grades are more likely similar to the normal cell than high grades, and also exhibit fast growth rates [24]. Moreover, high grades are also indicative of fast growth [24].

## 4. Diagnostic Strategies for CRC

### 4.1. Symptoms of CRC

The general symptoms in CRC patients are variations in bowel routines, rectal bleeding, blood in the stool, abdominal pain, cramping, weight loss, or fatigue. Typically, the early detection of colon cancer is vital for increasing the probability of treatment. Most people do not have severe symptoms in the early stages. The most common symptoms are rectal bleeding, dark stools, or blood in the stools, and changes in bowel habits, such as sudden constipation [4].

### 4.2. Methods of CRC Diagnosis

There are many methods that are utilized to diagnose the tumor in the early stage. Although making a CRC diagnosis using blood samples is very hard, and not used to detect CRC, there are many diagnostic methods available, as shown in Figure 6, including non-invasive and invasive methods. Advanced diagnostic methods, such as genetic and molecular characterizations of tumors, are available, but it remains combined with a poor diagnosis and very low rates of long-term survival [25]. In clinical diagnosis, these biomarkers, and molecular diagnosis, such as miRNA, are used to diagnose cancer with high precision and accuracy.

#### 4.2.1. Biochemical and Molecular Biomarkers

There are several types of biomarkers and biochemical molecules used in CRC diagnostics, such as proteins, tumor DNA, tumor-derived cells, miRNA, CEA, M2-PK, and microsatellite instability (MSI) (Table 2). These molecular compounds can be detected in bodily fluids, such as urine, cerebrospinal fluid, ascites, pleural effusion, blood, stool, and some liquid biopsies that disseminate malignant cells from any bodily fluid. This method of diagnosis is superior to fecal occult blood testing or fecal immunochemical testing in that it is quick, non-invasive, relatively inexpensive, and has higher sensitivity and specificity molecular biomarkers [26].

Circulating tumor cells are cancer cells present in the blood circulation or surround the tumor. It is one of the biomarkers that is used to detect or provide information about therapy methods [26].

#### 4.2.2. CRC Diagnosis via the Nano-Molecular Approach

In several cancer types, including those of the breast, lung, colon, pancreatic, prostate, gastric, ovarian, esophagus, and liver, the dysregulation of miRNAs, including miR-18a, miR-21, miR-155, miR-221, and miR-375, has been demonstrated [28]. Based on DNA markers found in blood and fecal samples, biomarkers for cancer detection are used. Fecal-based markers include fecal immunochemical test DNA. Additionally, numerous blood-based marker assays for proteins and glycoproteins, such as mucins, and cell-free DNA testing are available [29]. Several of the most critical genes needed for the transformation process may be limited, and work as signals to detect these cancers if they are located in the stool or tissue [30]. Some of the diagnostic chromosomes are presented in Table 3.

Circulating tumor DNA (ctDNA) is one of the methods to diagnose cancer patients. The high levels of cfDNA measured between 180 and 200 represent one of the biomarkers. This DNA or nucleosome is found in the blood due to the deregulation of caspases [26].

Although its concentration in the blood may increase and strokes, it is a highly reliable indicator for diagnosing and treating colorectal cancer, as well as in evaluating the effectiveness of treatment [26,31].

Circulating microRNA (c-miRNA) has recently been used as a non-invasive diagnostic method for CRC [32]. It plays a part in cell cycle regulation, dysregulation of miRNA function, and the activation of various illnesses like cancer. Usually, plasma has a higher concentration than serum. MiRNAs that are highly up- or down-regulated serve as clinical indicators for cancer. A total of 230 miRNAs that were distinctly expressed in the four stages of the change from benign large bowel mucosa to colorectal cancer were also identified [26,33].

SEPT9 methylation is one of the greatest blood markers to detect CRC. It is usually used as a method to define the CRC stage. In patients with early CRC stages between 0–I, the range could be from 57% to 64% [34]. The mSEPT9 positivity rate increased in advanced CRC cases [26].

Long non-coding RNA (lncRNA) is in all body fluids (including the blood, plasma, and urine). It has two classes; the first one is small non-coding RNAs, which have a length of around 200 nucleotides and could be microRNAs, miRNAs, and RNAs (siRNAs). The second type is long non-coding RNAs (lncRNAs), which are more than 200 nucleotides long and affect the cancer cells through several mechanisms. Several lncRNAs are associated with tumor stages and CRC development and are considered naturally antisense. However, this method takes a very long time and is expensive, and the virtue of the result is low [26].

#### 4.2.3. Chromosomal Biomarkers for CRC

Loss of heterozygosity (LOH) at chromosome 18q frequently occurs at a late stage during colon cancer development and is inversely associated with microsatellite instability (MSI). This LOH at chromosome 18q has been reported to predict shorter survival times in patients with colorectal cancer, whereas a MSI-high status has been associated with a superior prognosis [35]. The loss of heterozygosity at chromosome 18q in colon cancer is one of the biomarkers that can assist in making therapeutic decisions in patients found with CRC. This is because many studies have found that chemotherapy does not give complete recovery. Moreover, some studies have found that CRC patients with a P53 mutation (at chromosome region 17p13) need to be treated with chemotherapy compared with surgery [36].

#### 4.2.4. Fecal Metagenomics

A fecal test is usually the first test doctors ask to look for the blood in the sample. There are two test types, namely the guaiac-based fecal occult blood test, which is usually used to uncover any blood present in the feces (gFOBT), and the fecal immunochemical test (FIT), which is more precise, as it detects the hemoglobin in the sample [7] (Alana Biggers, 2021). Recently, fecal metagenomics meta-analysis studies using shot-gun metagenome data to diagnose CRC have been tested to confirm the applicability of the microbiome biomarkers across different inhabitants. There are 29 main species with major enhancement in the CRC metagenomes. Microbial signs were proven to be a current effective diagnostic method of CRC in its early stages, and they can also be reached to reduce the mortality associated with the disease [1]. In other studies, many studies showed that Parvimonas micra, *Solobacterium moorei*, and *Peptostreptococcus stomatitis* were found enriched in the microbiomes of the stools of patients. These were confirmed via PCR, with an enrichment of *Fusobacterium nucleatum* in stool and tissue samples from patients with colorectal carcinoma, and also by taxonomic studies, with a metagenomics analysis showing an enrichment of virulence-associated bacterial genes in the tumor microenvironment. Many studies of CRC patients compared with samples from controls reported a high prevalence of toxin genes expressed by some species, as described in Table 1, such as enterotoxigenic B and ETBF. This sample detected the bft gene in the mucosa, which encodes the bacterial toxin B. fragilis toxin (BFT); also, ETBF- and BFT-carrying strains were detected more often in the stool samples. The BFT-producing B fragilis was more prevalent in late-stage CRC [14]. Moreover, metabolomics is one of the biomarkers for early colon cancer detection. According to [1], recent studies have “showed that an altered state of the gut microbiota results in the reduction of the concentration of the SCFAs. These SCFAs usually obtained from carbohydrate fermentation in the colon are known to be essential components needed for the maintenance of gut homeostasis. However, in a state of gut microbiota dysbiosis, fermentation of these carbohydrates yields a much lower concentration of the SCFAs than they would in healthy states. Their quantification in stools has been said to possibly function as biomarkers for non-invasive diagnosis of various gut ailments” [1].

### 4.3. Other Biomarkers

Recent research has highlighted new gut biological markers for prognosis determination and CRC screening. However, there are many challenges to using biomarkers for clinical applications. First, most of the time, the sensitivity of the early stages of colon cancer is very weak; also, the presence of advanced adenomas. Moreover, the samples should be of very high quality, and the cost is too high [29]. However, the intrusive diagnostic methods are rather expensive and come with certain health concerns, including intraperitoneal bleeding, colon puncture, post-polypectomy bleeding, and the potential for infection. However, prior to making an invasive diagnosis, doctors typically advise utilizing a colonoscopy and a CT scan to thoroughly evaluate the entire colon. Once a problem is identified, surgery is then performed to fully assess the cancer stage to develop a better treatment strategy.

Furthermore, there are several studies involving some predictive biomarkers in CRC patients, such as anti-epidermal growth factor receptor therapy. This mutation affects 37% of patients and is most common in proliferative diseases. It is highly significant in CRC and is considered a predictive factor for anti-EGFR antibodies; moreover, it is used as a predictive factor therapy. The other biomarker is the potential toxicity of irinotecan. This factor inhibits apoptotic cell proliferation, and UGT1A1 is considered a biomarker of high importance for the potential toxicity of irinotecan. However, it does not concede in responding to treatments. Moreover, regarding the biomarkers of vascular endothelial growth factor-targeted therapy, some angiogenesis drugs have been introduced, such as bevacizumab, regorafenib, and aflibercept. However, there is no study showing their usefulness as a predictive factor.

### 4.4. Imaging

The diagnosis of CRC is made using a variety of imaging methods, including transrectal ultrasonography (TRU), MRI, and CTC. By evaluating the cancer’s size, degree of invasion into neighboring tissues, involvement of lymph nodes, and potential for dissemination to distant locations, imaging tools, including computed tomography (CT), magnetic resonance imaging (MRI), and positron emission tomography (PET) scans assist in staging the disease. Making treatment decisions is aided using this knowledge. These techniques might take advantage of surgical assessment for early screening and pre- or post-treatment comparisons. Oncologists and surgeons can more accurately design neoadjuvant therapy with the use of accurate imaging. For instance, it aids in choosing the best radiation field and dose, as well as in outlining the most efficient chemotherapy schedule. However, these are still less accurate in identifying small polyps, and could yield false-positive clinical stage results in CRC patients [2]. In some medical applications, nanoparticles are used, such as Fe_3_O_4_, α-Fe_2_O_3_, γFe_2_O_3_, PNA, and AuNPs, and there are different particles that have been outlined in Table 4 which can be considered as suitable candidates. To illustrate, the Fe_3_O_4_ (magnetite) α-Fe_2_O_3_ (hematite), and γ-Fe_2_O_3_ (maghemite) nanoparticles are options that can be applied in field emission scanning electron microscopy, X-ray photoelectron spectroscopy, energy dispersion X-ray spectroscopy, X-ray (XRD) photoelectron spectroscopy, Fourier transform infrared spectroscopy (FTIR), and cyclic voltammetry methods for cancer detection [37]. These techniques and the benefits of using some nanoparticles in imaging have been clarified in Table 4; also, Figure 7 illuminates the different active and passive targeting methods in the imaging.

### 4.5. Colonoscopy

Colonoscopy is the major diagnostic method used to detect colon cancer. It allows the doctor to see inside the colon using the camera or to take a tissue sample to send to a laboratory for analysis to clarify whether they are cancerous, precancerous, or noncancerous. It is very useful to determine the cancer stage, type, and area. In some cases, the tumor can be removed through the scope procedure. There are also different kinds of colonoscopy, including sigmoidoscopy, which can examine just the last section of the colon using a flexible tube with a light on it [7]. However, there are some health risks from colonoscopy, and it is classified as an invasive method. These risks can include bleeding from the biopsy that has been taken or from the tissue being removed, or sometimes tear or perforation in the colon or rectum wall during the colonoscopy [39]. As a screening, the doctors recommend a colonoscopy every five years, especially for people over 45 years and who have a family history of colon cancer.

### 4.6. Capsule Endoscopy

Capsule endoscopy can be used to evaluate the esophagus, stomach, small intestine, and colon. It is ingested just like any other capsule and travels through the esophagus into the stomach. It then passes through the pyloric sphincter into the duodenum, jejunum, and ileum [40].

### 4.7. CT Colonography

CT colonography is used to screen for cancers and other conditions affecting the colon. A previous study looked for significant growths, such as polyps, within the rectum and colon of patients. Polyps are growths on the colon’s lining that sometimes grow into cancers [41].

### 4.8. The Multi-Target Stool DNA Test

The multi-target fecal immunochemical test (FIT) and stool DNA test not only detect mutations associated with colorectal cancer, but also incorporate the FIT test to detect blood. The positive side of the test is that it can be performed at home, and it detects specific colorectal cancer-related mutations [42].

### 4.9. Diagnostic Strategy in Colorectal Cancer

Due to these advantages, neoadjuvant therapy is becoming the standard of care for an increasing number of tumor types. Currently, colon cancer patients are still routinely treated with up-front surgery, and neoadjuvant systemic therapy is still not yet a standard. Limitations to the widespread use of neoadjuvant therapy have included inaccurate radiological staging, concerns about tumor progression while undergoing preoperative treatment, rendering a patient incurable, and a lack of randomized data demonstrating benefits. Nevertheless, there is abundant interest in neoadjuvant chemotherapy, and a number of trials are underway. Initial follow-ups of the first phase III trial of neoadjuvant chemotherapy for colon cancer demonstrated tumor down-staging and suggested an improvement in disease-free survival with neoadjuvant chemotherapy [43].

## 5. Cancer Treatments

There are many types of treatment for patients diagnosed with colon cancer, as shown in Figure 8 below. The doctor gives different options for the treatment plan. The treatment also depends on the cancer stage and situation. In the very early stages, when it is a primary tumor, the treatment is easy and sometimes not very severe, and the patient does not need a long time to recover. However, when the cancer cells are found in different parts of the body at a very late stage, the treatment will be complicated and require a long time. Moreover, in some situations, the treatment plan will require two or three types of treatments. There are many difficulties in the treatment of CRC. Surgery is the cornerstone of treatment for CRC, while adjuvant chemotherapy is routinely applied to improve the prognosis of the patients [44]. However, chemoresistance is one of the major problems hindering the treatment of CRC. Since the existence of cancer stem cells (CSCs) leads to chemotherapy failure and tumor recurrence, targeting the CSCs could improve the therapeutic effectiveness in CRC [45]. Thus, an exploration of the molecules controlling the stemness of CRC will provide therapeutic targets for CRC.

### 5.1. Colon Surgery

In the early stages, and if the tumor is small and does not expand to other organs, local treatment is the first option for the patient. Local treatments mean surgery for colon cancer, and this is usually used for stages 0 and I. This surgery sometimes involves colonoscopy and sometimes by cutting the loop that has the cancer cell to remove all the affected area, and this is called a hemicolectomy, partial colectomy, or segmental resection. If the whole colon is removed [5] (ACSMECT, 2020), in some patients, the surgeon might reattach the healthy portion of the colon to the rectum. This procedure is called colostomy, and it needs to open in the abdominal wall the remove the waste [7]. However, new robotic techniques or enhanced visualization help improve the high-resolution, three-dimensional images of the surgical field, enabling surgeons to perceive fine details more clearly. This increased visibility makes it easier to spot important structures, cut them precisely, and inflict minimal damage to healthy tissues.

### 5.2. Chemotherapy

Chemotherapy is a systemic treatment, which means administering drugs through the blood or mouth to kill the tumor. This kind of treatment can be used at different times, and can also be combined with another therapy. Sometimes, this therapy is used after the surgery to make sure to avoid any remaining cancerous cells; moreover, sometimes it can be conducted before the surgery to shrink the size of the tumor to make it easier to remove, and it can also be used to control the tumor’s growth. The CRC chemotherapy drugs are capecitabine (Xeloda), fluorouracil, oxaliplatin (Eloxatin), 5-fluorouracil (5-FU), and irinotecan (Camptosar) (Table 5) [7]. Moreover, this therapy might be utilized for late-stage patients, when the tumor has spread all over their bodies, and in this case, it would be used to just help them live longer and feel better. The problem with this treatment is that it has many side effects. Chemotherapy usually works to track and attack cells that are quickly divided; that is why it is very harmful to hair and skin cells [5].

There are anticancer drugs that are available which can be used for the treatment of colorectal cancer alone or in combination with other drugs; for example (a) bevacizumab (Avastin) with fluorouracil as the first or second treatment, with a fluoropyrimidine and either irinotecan hydrochloride or oxaliplatin as the second treatment in patients whose disease has gotten worse after therapy that included bevacizumab in Stage III patients. (b) Cetuximab (Erbitux). In patients whose cancer has the EGFR protein and the wild-type KRAS gene, it is used with FOLFIRI combination chemotherapy as the first treatment, with irinotecan hydrochloride in patients whose cancer was treated with chemotherapy that included irinotecan hydrochloride but it either did not work or was no longer working, and alone in patients whose cancer did not respond to oxaliplatin and irinotecan hydrochloride or who cannot be treated with irinotecan hydrochloride. (c) Encorafenib (Braftovi). Encorafenib has been approved to be used with other drugs to treat patients whose cancer has a certain mutation in the BRAF gene. For colorectal cancer that has spread to other parts of the body, this drug is used with cetuximab in adults who have received previous treatment. (d) Ipilimumab (Yervoy): In adults and children aged 12 years and older, ipilimumab is used with nivolumab to treat metastatic microsatellite instability-high (MSI-H) or mismatch repair deficient (dMMR) cancer that got worse after treatment with a fluoropyrimidine, oxaliplatin, and irinotecan hydrochloride [46].

**Table 5 diseases-11-00148-t005:** List of nanotherapeutics for colon cancer.

Nanoparticles	Active Molecules	Application	Result	References
Mesoporous silica (MSN)	5-fluorouracil (5-FU)	Oral treatment of colon cancer	Better targeting of colon cancer cells	[47]
Dextran sulphate sodium	17-AAG	Orally administrated NP-PEG-FA/17-AAG	Better cancer therapy while reducing systemic exposure	[48]
Liposomes, cyclodextrin, nano gels, AuNPs, polymers, lipids, and micelles	Curcumin	Applied to colon cancer cell C26, HCT116	Enhances the internalization to inhibit tumor progression	[2]
Liposome-mesoporous silica NPs, lipid-core nanocapsules, and AuNPs	Resveratrol	CRC cells	Better cancer cell internalization and penetration	
PEGylated silica NPs	Genistein	CRC cells	Better anticancer activity	

### 5.3. The Targeted Therapies for CRC

CRC conventional treatments, such as radiotherapies, chemo, and biological agents, have improved cancer therapy; however, they produce many serious side effects, like drug resistance and toxicity to normal cells/tissues. The innovative strategies for targeted therapies, like drug delivery systems, diagnostic methods, and surgical practices, have led researchers to work and search more in these alternative therapies, including nanotechnology-based techniques [2]. Recent studies have focused on drug delivery via nanotechnology that is due to many factors, like the therapy time, drug efficacy, and side effects. Moreover, to reduce the side effects of chemotherapy, they have used nanotechnology to target the tumor without any effect on the other cells and organs. Many nanoparticles have succeeded in proving their effectiveness in delivering the drug today and eliminating cancer cells without any disarray [2]. Today, it is possible to administer anticancer medications to cancer cells without harming healthy or normal cells thanks to advancements in nanotechnology. The most ideal drug delivery systems to target cancer cells have been made possible using nanoparticles with a high drug distribution, improved bioavailability, good tumor penetration, and greater tissue half-life. The specific nanoparticle surface receptors help to improve colon cancer targeting due to binding ligands. Moreover, some studies think active-targeted nanotherapeutics can be translated into early-phase clinical trials [49]. Resveratrol (RES) (3,4′,5–trihydroxy stilbene) is a compound that can be extracted from certain plants, such as peanuts grapes, plums, and berries [50]. It was reported that RES suppressed metastasis and cell invasion in CRC cell lines via the suppression of the Wnt/β-catenin pathway [51], and [52] that RES restored EMT via the Akt/GSK-3β/Snail signaling pathway in CRC cells and tumors in nude mice. Treatment via RES elevated the epithelial expression of the marker E-cadherin and reduced the expression of mesenchymal markers, such as N-cadherin and Snail [53]. RES suppressed cell propagation, while stimulating apoptosis, and arrested the cell cycle at the G1 phase in CRC cells. In addition, Akt1 and Akt2 were detected as possible targets of RES. Moreover, it was found that Akt1/2 knockdown suppressed cell proliferation and colony synthesis in CRC cells. These effects were similar to those revealed by the treatment with RES [54]. RES is an effective anticancer product but is applied with significant limitations due to its low solubility in aqueous environments and low bioavailability. Therefore, many nano-drug carrier systems, like lipid nanomaterials, polymeric nanoparticles, micelles, liposomes, metals, such as silver and gold nanoparticles, and silica nanomaterials, have been used for RES loading [55,56]. These nano-formulations show distinctive physicochemical properties, like a high capacity of loading, protection of cargo, and penetrating deeply into the tumor, as well as safer and more suitable biocompatibilities. Liposomes are one of the previous lipid-based drug nano-delivery systems that include cholesterol and phospholipids with an internal hydrophilic core and an external hydrophobic bilayer of lipids [57]. This molecular system of the liposomes leads to an effective delivery and stimulates drug efficiency.

The role of the Wnt/β-catenin pathways have been studied in terms of cancer progression and development; it has been shown that Sec62 promotes the stemness and chemoresistance of human colorectal cancer through activating the Wnt/B-catenin pathway [58,59]. 

The liposome-based drug delivery system is also used in cancer therapy, as the liposome’s surface is used for PEGylation to promote receptor-mediated endocytosis. According to previous studies, long PEGylated liposomes enhance circulation half-life in vivo by lowering immune recognition, clearance, and non-specific serum protein absorption. Correspondingly, it reduces the onset of side effects. These liposomes used small molecules, like antibodies, peptides, proteins, and carbohydrates, as a targeting ligand or nano-carrier, as shown in Figure 9; these are functional imaging agents and cancer-specific ligands, as shown in Figure 10 [60]. There are two kinds of links, namely physical link and chemical link bonds, and both can be used for drug delivery, as shown in Figure 11.

### 5.4. Application of Phototherapeutic-Based Nanoparticles in Colorectal Cancer

To increase the efficiency of chemotherapy treatments, nowadays, they are being developed to encourage phototherapy and the characteristics of PSs. These drugs are more effective in destroying the tumor faster, as the affected part is exposed to different types of light with wavelengths, where the wavelength ranges from 600 to 1000 nanometers, including the long wave near-infrared (NIR) light [61], to penetrate the tissue and kill the cancerous tumor immediately. Moreover, recent studies have found that wavelengths with a high 1O2 yield are the best for irradiation. In some cases, there may be a need to increase the temperature to 42 degree Celsius, and this is called a photothermal action [62].

When a photosensitizer (PS) is combined with the cancerous tissues or tumor location during photothermal therapy (PTT), the energy of the targeted laser wavelength is transformed into heat energy, thermally destroying the cancerous tissue or tumor cells. PTT provides observable advantages over conventional cancer therapy due to its low invasiveness, decreased cell toxicity, and high drug metabolism. Surprisingly, PTT is a successful therapeutic option for CRC patients [62,63]. The examination of cancer using imaging techniques, such as fluorescence imaging, photoacoustic imaging, nuclear magnetic resonance imaging, and nuclear imaging, provided the basis for early cancer detection, as single phototherapy has several restrictions, especially regarding deep tumor penetration [62,63]. There are different types, shapes, and sizes of nanoparticles that are used as drug carriers to carry photosensitive molecules under an optical therapy for the improvement of drug penetration inside cancer tissues or tumors [54,63,64]. The molecular mechanism of PTT is the selective killing of the cancer cells where the various internal organelles of the cells, such as mitochondria [65,66], lysosomes [67,68], cell membranes [69], nuclear and genetic material [70,71], and blood vessels [72,73], are damaged, which can lead to cancer cell death.

Perioperative treatment combined with radical resection is the major approach to cure non-metastatic colon cancer. A precise evaluation and perioperative treatment would probably improve the R0 resection rate, recurrence-free survival, and overall survival of colon cancer patients. Recently, individualized treatment has been the mainstream due to the development of molecular pathology and multi-disciplinary therapy. The indications and course of perioperative treatment and preoperative neoadjuvant therapy of colon cancer are still in intense discussion. Firstly, the various reactions of adjuvant therapy to stage II colon cancer is caused by patients’ heterogeneity. Choosing a stratified treatment for these patients according to their clinical and molecular pathological features is the future. Secondly, the clinical trial of an adjuvant chemotherapy course for stage III colon cancer is under progress [74].

### 5.5. Female Hormones: Prospective Avenues for CRC as a Nano Therapy

Globally, a higher rate of CRC is reported in males than females due to the protective activity of sex hormones in the progression of the illness. Experimental research promoted the function played by estrogens and their receptors in the inception and progress of CRC in that the defensive role of estrogen is mediated via ERβ. As mentioned, in the postmenopausal era, HRT, in addition to consuming soy, decreases the occurrence of CRC. In the Women’s Health Initiative trial, HRT usage in women in the postmenopausal phase decreased the chance of CRC by 56%. Another meta-analysis study revealed that consuming soy by women decreased the chance of risk of disease by 21%. In tumorigenesis, ERβ expression has been lost in colonocytes. However, HRT, estrogen ligands, or products of soy, spend their activity to avoid a reduction in ERβ expression. Consequently, in the adenoma-to-carcinoma progress, the timing of HRT’s intervention is a key determinant of the benefit. The defensive actions of estrogen are dependent on some molecular secondary types. Thus, success in developing estrogen modulators for the protection of CRC is based on the detection of the liable CRC population [75]. Estrogen receptor (ERβ) is the dominant estrogen receptor type that is expressed in an epithelial layer in either a normal state or in a tumor in the colon. Nevertheless, in the malignant progression of colon cancer, the expression of ERβ is missing, which indicates that the signaling of estrogen might partly participate in the progression of the illness. However, estrogenic hormones may exhibit anticancer activity via the elective stimulation of pro-apoptosis signaling mediated by ERβ, the suppression of inflammation signals, as well as the modification microenvironment of the tumor. It has been reported that the pathway of estrogen has been used as a potential treatment route for CRC, and showed the mechanisms that estrogen can mediate through the protective path versus CRC tumorigenesis at the cellular and molecular levels, which made estrogen a possible avenue for targeted treatment [76]. The application of hormonal replacement therapy (HRT) by post-menopausal females has revealed a connection with a decreased danger of CRC, as many meta-analysis works in different countries have been carried out to systematically estimate the survival merit using HRT for CRC patients. They showed that the use of HRT is associated with a decreased risk in CRC-specific and total death rates in CRC patients [77,78]. Furthermore, in many population-based studies in Sweden, it has been confirmed that applying HRT after a diagnosis of CRC was not only connected with a decreased risk of cancer-related mortality but also with all-cause mortality [51,79,80].

Additionally, another sex female hormone, progesterone, has been associated with a reduction in CRC risk and positive anticipation. Low expression levels of progesterone and progesterone receptor (PGR) have been markedly connected with CRC. Additionally, progesterone inhibited the proliferation of the CRC cell line in vitro by interrupting the cell cycle and inducing apoptosis. Additionally, new in vivo investigations on the inhibitory effect of progesterone on tumor growth were conducted. Progesterone also increased the rate of DNA damage-inducible protein (GADD45) and growth arrest, which, in turn, stimulated JNK signaling. Progesterone enhanced the JNK pathway’s action through GADD45 to inhibit propagation via cell cycle arrest and promote apoptosis, hence inhibiting the advancement of CRC. Progesterone and PGR may, therefore, act as suppressing variables to reduce the likelihood of CRC [81]. Combination therapy with estrogen and progesterone was found to increase the expression of caspase 3, cleaved PARP, and cleaved caspase 8 in tumors, while decreasing the expression of the full name (PCNA), indicating the extrinsic pathway of apoptosis [82,83]. Moreover, a greater expression of ERβ was observed in the tumors. They concluded that the effect of both estradiol and progesterone hormones is important for decreasing proliferation and increasing apoptosis in CRC, mostly via the activation of ERβ.

### 5.6. Adjuvant Therapy and Neoadjuvant Therapy in Cancer

In patients with stage III colon cancer (CC), adjuvant chemotherapy with the combination of oxaliplatin with fluoropyrimidine (FOLFOX or CAPOX) is a standard of care. The duration of treatment can be reduced from 6 months to 3 months, depending on the regimen, for patients at a low risk of recurrence, without a loss in effectiveness and allowing for a significant reduction in the risk of cumulative sensitive neuropathy [74]. Neoadjuvant systemic therapy has many potential advantages over up-front surgery, including tumor down-staging, early treatment of micro-metastatic disease, and providing an in vivo test of tumor biology. Due to these advantages, neoadjuvant therapy is becoming the standard of care for an increasing number of tumor types. Currently, colon cancer patients are still routinely treated with up-front surgery, and neoadjuvant systemic therapy is not yet standard. Limitations preventing a widespread use of neoadjuvant therapy have included inaccurate radiological staging, concerns about tumor progression while undergoing preoperative treatment, rendering a patient incurable, and a lack of randomized data demonstrating benefit. Nevertheless, there is abundant interest in neoadjuvant chemotherapy, and a number of trials are underway. An initial follow-up of the first phase III trial of neoadjuvant chemotherapy for colon cancer demonstrated tumor down-staging and suggested an improvement in disease-free survival with neoadjuvant chemotherapy [43].

### 5.7. Future Prospects of Colorectal Treatment

Heparin-polyethyleneimine (HPEI) nanoparticles were used to deliver plasmids expressing mouse survivin-T34A (ms-T34A) to treat colon cancer C-26 carcinoma in vitro and in vivo. According to the in vitro studies, HPEI nanoparticles could efficiently transfect the pGFP report gene into the C-26 cells, with a transfection efficiency of 30.5% ± 2%. Intratumoral injections of HPEI nanoparticle-mediated ms-T34A significantly inhibited the growth of subcutaneous C-26 carcinoma in vivo through the induction of apoptosis and the inhibition of angiogenesis [84]. In another study, it was found that that the gef gene induced a marked decrease in cell viability (50% in 24 h) in T-84 cells through cell death via apoptosis. Interestingly, when gef gene expression was combined with several drugs of choice in the clinical treatment of colon cancer (5-fluorouracil, oxaliplatin, and irinotecan), a strong synergistic effect was observed with an approximated 15–20% enhancement of the anti-proliferative effect [85]. In addition, reactivation of survivin expression is involved in carcinogenesis and angiogenesis in colon cancer. Regarding the therapeutic effect of the adeno-associated virus (AAV)-mediated survivin mutant (Cys84Ala) on colon cancer, it was reported that the transduction of colon cancer cells with rAAV-Sur-Mut(Cys84Ala) inhibited cell proliferation and induced apoptosis and mitotic catastrophe in vitro. rAAV-Sur-Mut(Cys84Ala) sensitized colon cancer cells to chemotherapeutic drugs. Intratumoral injections of rAAV-Sur-Mut(Cys84Ala) significantly induced apoptosis and mitotic catastrophe and inhibited angiogenesis and tumor growth in a colon cancer xenograft model in vivo [86]. Recently, the roles of the miR-34a and miR-34b/c encoding genes were uncovered in the context of colon cancer therapy [87]. Dysbiosis of the gut microbiome has been associated with the development of colorectal cancer (CRC). The gut microbiota is involved in the metabolic transformations of dietary components into oncometabolites and tumor-suppressive metabolites that, in turn, affect CRC development. The alpha-bug hypothesis suggested that oncogenic bacteria, such as enterotoxigenic *Bacteroides fragilis* (ETBF), induce the development of CRC through direct interactions with colonic epithelial cells and alterations of microbiota composition at the colorectal site. *Escherichia coli*, *E. faecalis*, *F. nucleatum*, and *Streptococcus gallolyticus* showed a higher abundance, whereas *Bifidobacterium*, *Clostridium, Faecalibacterium*, and *Roseburia* showed a reduced abundance in CRC patients [88]. The application of the exogenous *Lactobacillus reuteri* restricts colon tumor growth, increases tumor reactive oxygen species, and decreases protein translation in vivo [89].

There are also some other systemic treatments, such as immunotherapy radiation and target therapy. Immunotherapy is usually used for the advanced stages to improve the immune system to detect and kill cancer cells. Radiation therapy uses sources of powerful energy, like X-rays and protons, to destroy cancer cells. It is utilized to shrink the size of large cancer cells. Radiation can also be combined with chemotherapy [39].

## 6. Conclusions

Colon cancer is the third cancer cause of death. It is a serious disease, but it is highly treatable when caught early. Regular screenings are the best way to detect it early, and there are many treatments available for those who are diagnosed. This review is like an overview of colon cancer cells, and it covers many sides of this disease. It started from the risk factor to the treatment, and focused on different treatment kinds. For that, we found that this type of cancer is much more severe and can be malignant and migrate to many parts of the body. However, the best way to prevent colon cancer is to undergo regular screenings. These tests can detect cancer early when it is most treatable. Colon cancer screening should especially be conducted on high-risk individuals, like the elderly or those with a complex medical or genetic history. Regular screening via colonoscopy has many side effects and is not comfortable. The new screening technique utilizing nanotechnology should solve this problem and make it more secure. Moreover, the new nano application helps to improve the treatment and diagnosis of CRC.

## Figures and Tables

**Figure 1 diseases-11-00148-f001:**
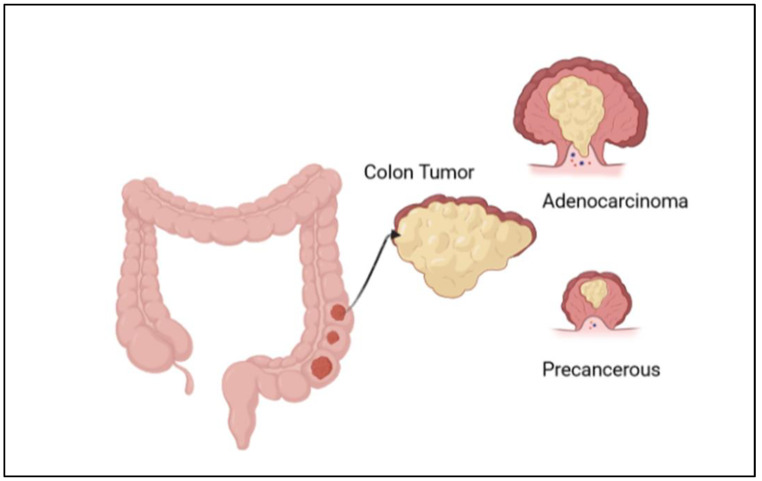
Illustration of the development of colon cancer, with the formation of adenocarcinoma and a precancerous stage.

**Figure 2 diseases-11-00148-f002:**
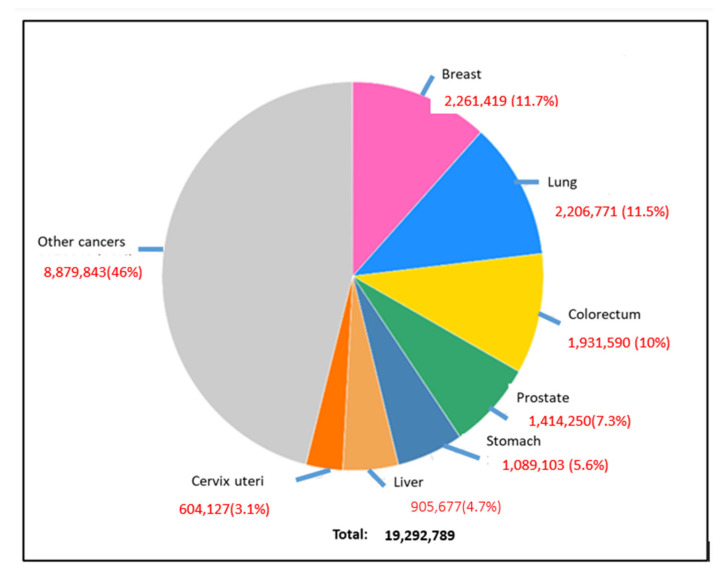
Global occurrence of different types of cancers in the human population, where breast cancer, lung cancer, and colorectal cancer are prominent [3].

**Figure 3 diseases-11-00148-f003:**
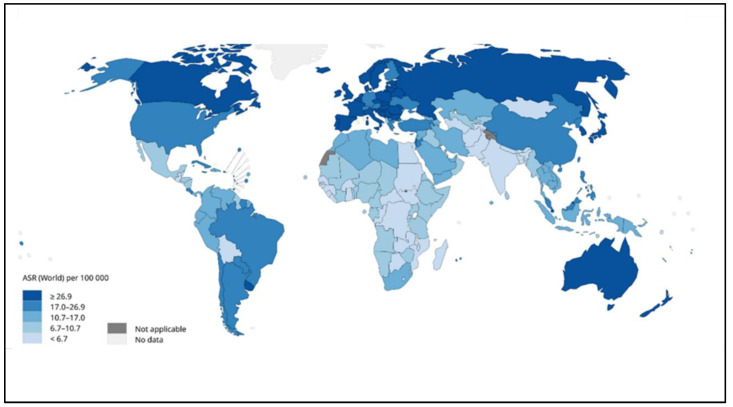
Global estimated age-standardized incidence rates of colon cancer in 2020 [12].

**Figure 4 diseases-11-00148-f004:**
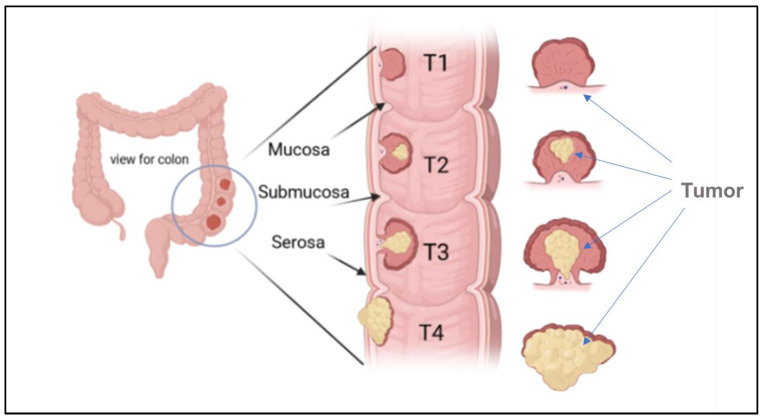
Illustration of the growth of a tumor through the mucosa layer of the colon, and different CRC stages based (T1, T2, T3, and T4) on the TNM system.

**Figure 5 diseases-11-00148-f005:**
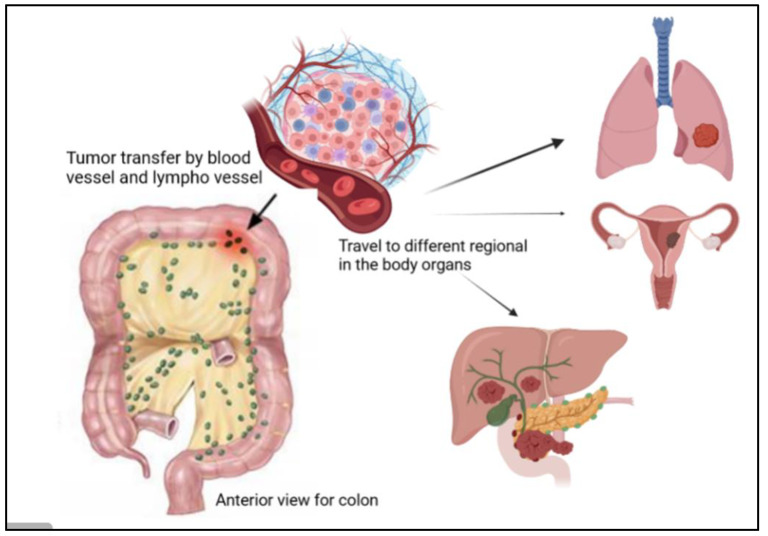
Illustration of the presence of cancer tumor cells in the colon, where they can travel to different regions of the human body, such as the lungs, liver, and ovaries.

**Figure 6 diseases-11-00148-f006:**
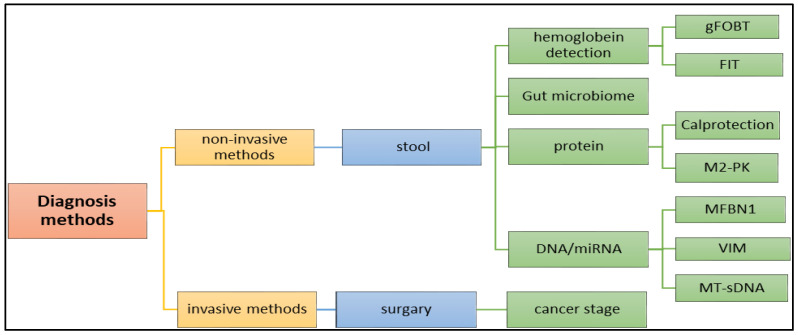
Application of various methods, including (1) invasive and (2) non-invasive methods in the diagnosis of the colon cancer.

**Figure 7 diseases-11-00148-f007:**
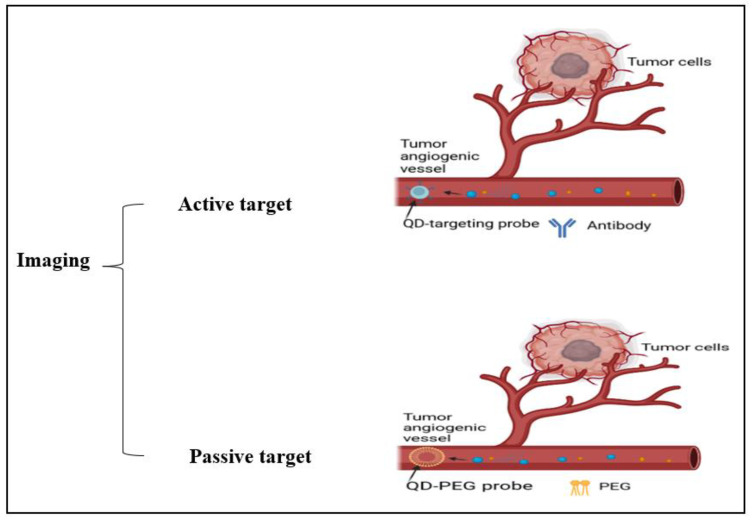
Illustration of cancer cell imaging using active targeting (antibodies) and passive targeting (PEG) methods.

**Figure 8 diseases-11-00148-f008:**
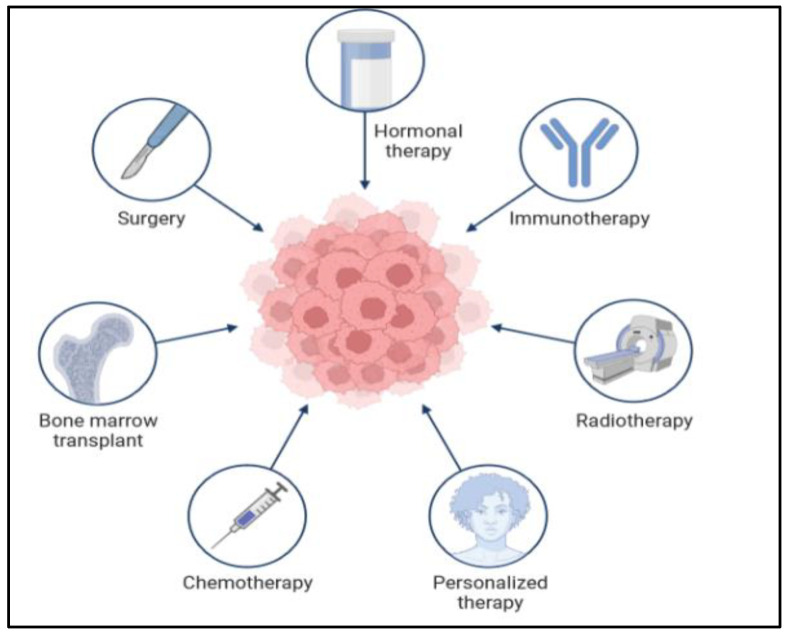
Application of different methods in the treatment of colon cancer.

**Figure 9 diseases-11-00148-f009:**
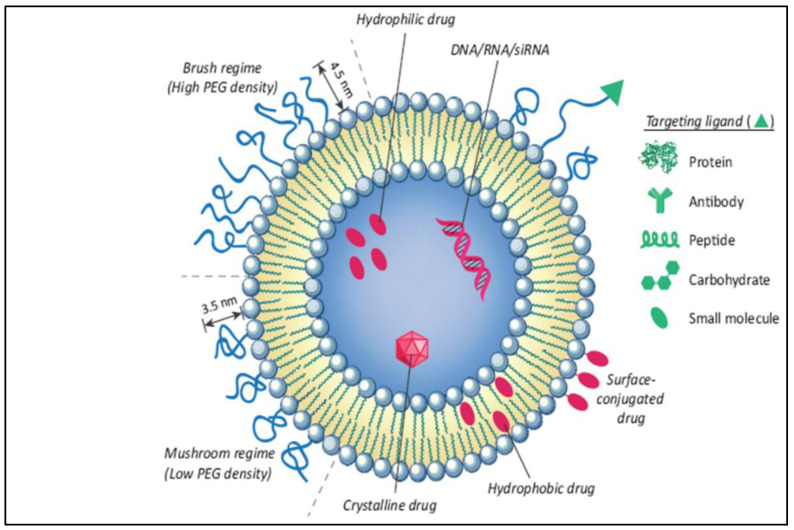
Illustration of the conjugation of an anticancer drug with a nano-formulation for effective liposomal drug delivery [60].

**Figure 10 diseases-11-00148-f010:**
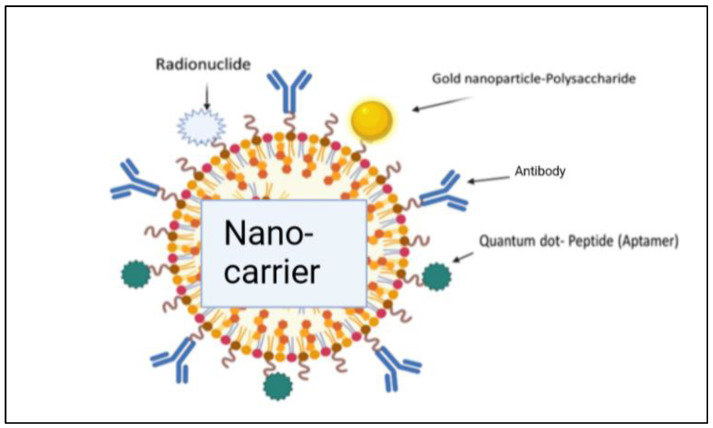
Illustration of the conjugation of a drug with nano-carriers for effective delivery of the drug molecules [2].

**Figure 11 diseases-11-00148-f011:**
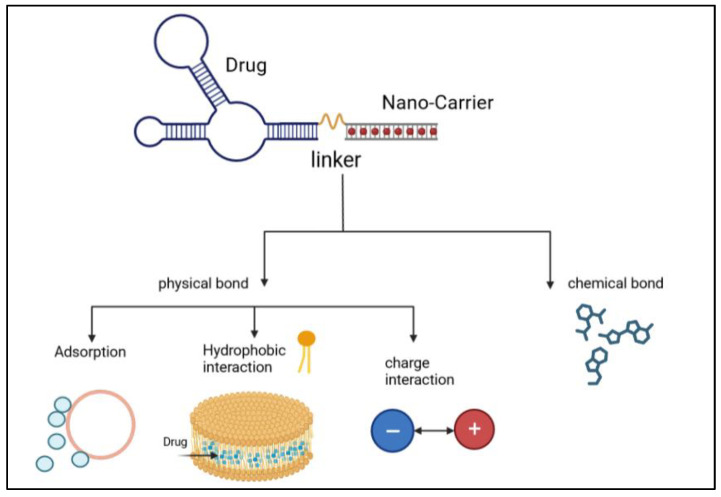
Illustration of the process of linking of an anticancer drug with nanocarriers via physical bonds and chemical bonds [2].

**Table 1 diseases-11-00148-t001:** CRC risk factors.

Kind	Factor	Example
History	Genetic from family history and the medical history of the patient	Number of relatives Inflammatory bowel disease Cancer treatments Have diabetes
Lifestyle	Lifestyle for the patient in the active and diet surgery	Overweight Physical activity and exercise Eat high amounts of red meat Drinking alcohol and smoking

**Table 2 diseases-11-00148-t002:** Molecular diagnostic methods.

Molecular Marker	Description	Diagnostic Methods	Reference
CEA (carcinoembryonic antigen)	Protein in the blood.	Measured using a blood test. A high level of CEA indicates the presence of colon cancer.	[27]
Calprotectin	Protein in the stool.	A non-invasive screening tool. A higher level than normal in the stool can indicate the presence of CRC.	
M2-PK	This is an enzyme that is often elevated in the blood of patients with colon cancer.	Measured using a blood test. A high level of M2-PK indicates the presence of colon cancer.	
S100A12	A protein in the blood.	Measured using a blood test. A high level of S100A12 can indicate the presence of colon cancer.	
Microsatellite instability (MSI)	This is a genetic change.	This genetic testing of tumor tissue. The presence of MSI can indicate a higher likelihood of colon cancer.	
KRAS mutation	This is a genetic changepresent in around 40% of colon cancer cases.	Diagnosed through genetic testing of tumor tissue. The presence of a KRAS mutation can indicate a higher likelihood of colon cancer.	

**Table 3 diseases-11-00148-t003:** Numbers of the most critical genes that are needed and their locations [30].

Specimen	Chromosomal Location
Stool DNA	5q21, 17p13, and 12p
Tissue	17p13, 3p22, 2q31–33, 2p21, 18q21, 20q11, 18q21, 2p21, 7p22, and 3p21
Colonic effluent and colonic DNA	12p

**Table 4 diseases-11-00148-t004:** Nanoparticles that are used to improve imaging.

Nanoparticles	Imaging Technique	Benefit	Result	Reference
Peanut agglutinin (PNA)	Florescence endoscopy	Early-stage small-sized CRC tissue	Better imaging of the CRC cell’s mucous	[2]
maghemite Fe_2_O_3_ and γ-Fe_2_O_3_	SERS technique	Crucial target for diagnosing CRC		
AuNPs	Raman spectroscopy (SERS)	Through the RS it differentiates between normal cells and cancerous serum	Better visibility and detection of cancerous serum	
Quantum dots (QDs)	Florescence	Emit from visible to infrared wavelengths upon excitation	Better cancer imaging	
Raman scattering (RS) and fiber optic probes		Hollow organs and diagnose CRC	Single-cell-based technique with better cancer imaging	
H2O2	Florescence	Circulating tumor cell detection	Better tumor targeting and imaging	[38]
Element barium	X-ray	Help to improve the quality of the X-ray images	Barium solution coats the lining of the colon	[7]

Element barium X-rays help to improve the quality of the X-ray images. The barium solution coats the lining of the colon [7].

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
