# Peer review of "Progress and Perspectives in Colon Cancer Pathology, Diagnosis, and Treatments"

_diseases, 2023, doi:10.3390/diseases11040148_

Round 1

Reviewer 1 Report

thank you for allowing me to review this literature. 

The title gives the impression that the authors are going to deal with colon cancer, but the abstract mentions radiotherapy, so I deduced that they were referring to colorectal cancer. 

At the end of the manuscript, I can't figure out what the authors wanted to develop: the prospects of nanotechnologies in colorectal cancer? if so, then the whole manuscript needs to be revised, as this review is incomplete in many respects: 

diagnostic strategy in colorectal cancer: the place of imaging in rectal cancer 

neoadjuvant therapeutic strategy in advanced colon cancer and locally advanced subperitoneal rectal cancer 

the place of watch and wait in rectal cancer

the contribution of new robotic technologies in colorectal surgery

total removal of the mesocolon 

advances in adjuvant treatment.

i therefore suggest rewriting this review, focusing on the place and interest of nanotechnologies and their prospects in colorectal cancer. 

Author Response

  1. The title gives the impression that the authors are going to deal with colon cancer, but the abstract mentions radiotherapy, so I deduced that they were referring to colorectal cancer. At the end of the manuscript, I can't figure out what the authors wanted to develop: the prospects of nanotechnologies in colorectal cancer? if so, then the whole manuscript needs to be revised, as this review is incomplete in many respects

Response: Thank you for your comment. Abstract has been revised accordingly.

Worldwide, colon cancer is the third most frequent malignancy and the second most common cause of death. Although it can strike anybody at any age, colon cancer mostly affects elderly people. Small, non-cancerous cell clusters inside the colon, commonly known as polyps, are typically where colon cancer growth starts. But over time, if left untreated, these benign polyps may develop into malignant tissues and develop into colon cancer. For the diagnosis of colon cancer with routine inspection of the colon region for polyps, several techniques including colonoscopy and cancer scanning are used. In the case of identification of the polyps in the colon area, efforts are being taken to surgically remove the polyps as quickly as possible before they become malignant. If the polyps become malignant, then colon cancer treatment strategies such as surgery, chemotherapy, targeted therapy, and immunotherapy are applied to the patients. Despite the recent improvements in diagnosis and prognosis, the treatment of colorectal cancer (CRC) remains a challenging task. The objective of this review is to discuss, how CRC is initiated, and its various developmental stages, pathophysiology, and risk factors, also, to explore the current state of colorectal cancer diagnosis and treatment, as well as recent advancements in the field such as new screening methods and targeted therapies. We will examine the limitations of current methods and discuss the ongoing need for research and development in this area. While the topic may be serious and complex, we hope to engage and inform our audience on this important issue.

  1. Diagnostic strategy in colorectal cancer: the place of imaging in rectal cancer neoadjuvant therapeutic strategy in advanced colon cancer and locally advanced subperitoneal rectal cancer the place of watch and wait in rectal cancer

Response: Thank you for the comment. Due to these advantages, neoadjuvant therapy is becoming the standard of care for an increasing number of tumor types. Currently, colon cancer patients are still routinely treated with up-front surgery, and neoadjuvant systemic therapy is not yet standard. Limitations to widespread use of neoadjuvant therapy have included inaccurate radiological staging, concerns about tumor progression while undergoing preoperative treatment rendering a patient incurable, and a lack of randomized data demonstrating benefit. Nevertheless, there is abundant interest in neoadjuvant chemotherapy, and a number of trials are under way. Initial follow up of the first phase III trial of neoadjuvant chemotherapy for colon cancer demonstrated tumor down-staging and suggested an improvement in disease-free survival with neoadjuvant chemotherapy [Body et al., 2021]

- Body A, Prenen H, Latham S, Lam M, Tipping-Smith S, Raghunath A, Segelov E. The Role of Neoadjuvant Chemotherapy in Locally Advanced Colon Cancer. Cancer Manag Res. 2021 Mar 17;13:2567-2579. doi: 10.2147/CMAR.S262870. PMID: 33762848; PMCID: PMC7982559.

  1. The contribution of new robotic technologies in colorectal surgery

Thank you for your comment. We have revised it. The information has been added.

  1. Total removal of the mesocolon 

Response: Thank you for your comment. We have revised it.

  1. Advances in adjuvant treatment.

Response: Thank you for your comment. We have revised it.

  1. Therefore suggest rewriting this review, focusing on the place and interest of nanotechnologies and their prospects in colorectal cancer. 
  2. Response: Thank you for your comment. We have revised the manuscript accordingly.

Reviewer 2 Report

In this review, the authors discussed how CRC is initiated and summarized the current state of colorectal cancer diagnosis and treatment. They also discussed the limitations of current methods and discuss the ongoing need for research and development in this area. The topic is interesting and meaningful. The manuscript is well organized. However, some issues should be addressed.

1. The keynotes of this review should be specifically presented in the introduction part. This will be helpful for the readers.

2. It’s suggested to add a table to summarize the risk factors.

3. It’s suggested the discussion part. What are the difficulties in CRC diagnosis and treatment? What can we do in the next basic or clinical studies? These should be discussed.

4. It’s suggested to cite the related papers (PMID: 33858476).

5. Which software did the authors use to create the figures? This should be described.

6. The figure legends are too simple.

The english grammar should be examined and some typo errors should be corrected.

Author Response

In this review, the authors discussed how CRC is initiated and summarized the current state of colorectal cancer diagnosis and treatment. They also discussed the limitations of current methods and discuss the ongoing need for research and development in this area. The topic is interesting and meaningful. The manuscript is well organized. However, some issues should be addressed.

  1. The keynotes of this review should be specifically presented in the introduction part. This will be helpful for the readers.

      Response: Thank you for your comment. We have revised it.

  1. It’s suggested to add a table to summarize the risk factors.

Response: We have discussed the risk factor in the revised manuscript. The incidence of colon and rectal cancer increases with age, but about 40% of cases are diagnosed after the age of 75 [86, 87]. About 40% of patients are diagnosed with the disease at stage I-II, 40% at stage III and 20% at stage IV [88, 89]. Patients with a history of colonic polyps and patients with a history of colon or rectal cancer or colonic polyps in close relatives are at increased risk [90, 91]. Inflammatory diseases of the intestine, i.e. ulcerative colitis and Crohn's disease , increase the risk of colon cancer [92]. It is believed that the probability of colon cancer in such patients is around 15-20% after 30 years, and the risk is possibly higher with ulcerative colitis [92]. Tobacco smoking appears to increase both the risk of colon and rectal cancer as well as the death rate caused by it [93] physical activity,[94] daily aspirin use [95] and vitamin D with calcium appear to have a protective effect [96]. Increased consumption of vegetables and fruits and reduced meat consumption seem to potentially reduce the risk of cancer, especially in the left part of the colon [86].

  1. Haraldsdottir S, Einarsdottir HM, Smaradottir A, Gunnlaugsson A, Halfdanarson TR. Krabbamein í ristli og endaþarmi [Colorectal cancer - review]. Laeknabladid. 2014 Feb;100(2):75-82. Icelandic. doi: 10.17992/lbl.2014.02.531. PMID: 24639430.
  2. Howlader N, Noone AM, Krapcho M, Neyman N, Aminou R, Altekruse SF, et al. SEER Cancer Statistics Review, 1975-2009 (Vintage 2009 Populations), National Cancer Institute. Bethesda, MD, seer.cancer.gov/csr/1975_2009_pops09 /, based on November 2011 SEER data submission, posted to the SEER web site, April 2012.
  3. Siegel R, Naishadham D, Jemal A. Cancer statistics, 2012. CA Cancer J Clin 2012; 62: 10-29.
  4. Alexiusdottir KK, Moller PH, Snaebjornsson P, Jonasson L, Olafsdottir EJ, Bjornsson ES, et al. Association of symptoms of colon cancer patients with tumor location and TNM tumor stage. Scand J Gastroenterol. 2012; 47: 795-801.
  5. Winawer SJ, Zauber AG, Gerdes H, O'Brien MJ, Gottlieb LS, Sternberg SS, et al. Risk of colorectal cancer in the families of patients with adenomatous polyps. National Polyp Study Workgroup. N Engl J Med 1996; 334: 82-7.
  6. Atkin WS, Morson BC, Cuzick J. Long-term risk of colorectal cancer after excision of rectosigmoid adenomas. N Engl J Med 1992; 326: 658-62.
  7. Farraye FA, Odze RD, Eaden J, Itzkowitz SH. AGA technical review on the diagnosis and management of colorectal neoplasia in inflammatory bowel disease. Gastroenterology 2010; 138: 746-74, 774 and 741-744; quiz e712-743.
  8. Botteri E, Iodice S, Bagnardi V, Raimondi S, Lowenfels AB, Maisonneuve P. Smoking and colorectal cancer: a meta-analysis. JAMA 2008; 300: 2765-78.
  9. Boyle T, Keegel T, Bull F, Heyworth J, Fritschi L. Physical Activity and Risks of Proximal and Distal Colon Cancers: A Systematic Review and Meta-analysis. J Natl Cancer Inst 2012; 104: 1548-61.
  10. Rothwell PM, Wilson M, Price JF, Belch JF, Meade TW, Mehta Z. Effect of daily aspirin on risk of cancer metastasis: a study of incident cancers during randomized controlled trials. Lancet 2012; 379: 1591-601.
  11. Park Y, Leitzmann MF, Subar AF, Hollenbeck A, Schatzkin A. Dairy food, calcium, and risk of cancer in the NIH-AARP Diet and Health Study. Arch Intern Med 2009; 169: 391-401.

      Response: Thank you for your comment. We have revised it

  1. It’s suggested the discussion part. What are the difficulties in CRC diagnosis and treatment? What can we do in the next basic or clinical studies? These should be discussed.

Response: There are many difficulties in the treatment for CRC. Surgery is the cornerstone of treatment for CRC while adjuvant chemotherapy is routinely applied to improve the prognosis of the patients [Siegel et al., 2020]. However, chemoresistance is one of the major problems hindering the CRC treatment. Since the existence of cancer stem cells (CSCs) leads to chemotherapy failure and tumor recurrence, targeting the CSCs could improve the therapeutic effectiveness in CRC [Ramos et al., 2017]. Thus, exploration of molecules controlling the stemness of CRC will provide therapeutic targets for CRC.

  • Siegel RL, Miller KD, Goding Sauer A, Fedewa SA, Butterly LF, Anderson JC, Cercek A, Smith RA, Jemal A. Colorectal cancer statistics, 2020. CA Cancer J Clin. 2020
  • Ramos EK, Hoffmann AD, Gerson SL, Liu H. New opportunities and challenges to defeat cancer stem cells. Trends Cancer. 2017;3(11):780–796.
  1. It’s suggested to cite the related papers (PMID: 33858476).

Response: We have citated the related papers

1-Liu X, Su K, Sun X, Jiang Y, Wang L, Hu C, Zhang C, Lu M, Du X, Xing B. Sec62 promotes stemness and chemoresistance of human colorectal cancer through activating Wnt/β-catenin pathway. J Exp Clin Cancer Res. 2021 Apr 15;40(1):132. doi: 10.1186/s13046-021-01934-6. PMID: 33858476; PMCID: PMC8051072.

2- Nusse R, Clevers H. Wnt/beta-catenin signaling, disease, and emerging therapeutic modalities. Cell. 2017;169(6):985–999. doi: 10.1016/j.cell.2017.05.016.

3- Shen C, Wang D, Liu X, Gu B, Du Y, Wei FZ, Cao LL, Song B, Lu X, Yang Q, et al. SET7/9 regulates cancer cell proliferation by influencing beta-catenin stability. FASEB J. 2015;29(10):4313–4323. doi: 10.1096/fj.15-273540.

Which software did the authors use to create the figures? This should be described.

Response: We have used biorender software https://app.biorender.com/ to create the figures

  1. The figure legends are too simple.

Response: The figure legends has improved:

Figure 1: Illustration of development of colon cancer with formation of adenocarcimoa and precancerous stage

Figure 2: Global occurrence of different types of cancers in human population where breast cancer, ling cancer and colorectal cancer are prominent.

Figure 5: Illustration of presence of cancer tumor cells in the colon, where they can travel to different region of the human body such as lung, liver and ovaries.

Figure 6: Application of various methods (1) invasive and (2) non-invasive in the diagnosis of the colon cancer

Figure 7: Illustration of cancer cell imaging by using active targeting (Antibodies) and passive targeting (PEG) methods are applied

Figure 8: Application of different methods in the treatment of colon cancer

Figure 9: Illustration of conjugation of anticancer drug with nano-formulation for effective liposomal drug delivery

Figure 10: Illustration of conjugation of drug with nano-carriers for effective delivery of the drug molecules

Figure 11: Illustration of process of linking of anticancer drug with nanocarriers by physical bonds and chemical bonds

Reviewer 3 Report

The authors in this review discuss on Colon cancer that is the third most frequent cancer and the second greatest cause of death worldwide. Despite advances in diagnosis and treatment, colorectal cancer (CRC) remains difficult to diagnose and treat, prompting this review to investigate its initiation, developmental stages, pathophysiology, risk factors, current diagnostics, treatment, and emerging advancements, emphasizing the importance of ongoing research and awareness on this critical issue. 

The authors have addressed all of the important limitations in CRC diagnosis and treatment options that are now available. Although there are now limitations to this, I feel that with the advancement of research, such restrictions can be addressed, particularly with regard to non-invasive means of diagnostic and treatment options. It would be interesting to see if artificial intelligence may be employed for CRC screening in the future. This review is sure to grab the readers' interest. Here are my thoughts.

Figure 6: check the spelling of “surgery”

5.2 Chemotherapy: Mention the stage specific drugs such as the combination of 5FU, leucovorin, and oxaliplatin may be prescribed for stage III. 

The use of phototherapeutic-based nanoparticles in colorectal cancer treatment is a unique and promising method, but like many medical therapies, may have negative effects. It is critical to recognize that the possible adverse effects might vary based on the type of nanoparticles employed, the patient's particular features, and the treatment procedures. Please include a discussion on this. 

Please comment on this issue.  The complexity of hormone regulation inside the human body is one of the key problems. Female hormones, such as estrogen and progesterone, have complex interactions that vary widely from person to person. Because of these differences, developing a one-size-fits-all nanotherapy method for CRC can be difficult.

Please comment on this issue. Hormonal therapy may have side effects and hazards that must be carefully considered, including the possibility of deleterious effects on other physiological systems. As a result, while the notion of using female hormones for CRC nano treatment is exciting, it requires more study and refining to solve these deficiencies and fully realize its therapeutic potential.

Please specify the “change” in the line 199. Instead, more specific words can be used such as gain, LOH etc. Include discussion on chromosomal instability under chromosomal biomarkers for CRC.

The gut microbiota is extremely diverse, with individual differences. Because of this variety, as well as the existence of non-cancer-related microbiota, identifying particular microbial markers linked with CRC might be difficult. While it can identify some CRC-associated microbial signatures, can it reliably identify all instances, which might result in findings that are either falsely positive or negative?

Under methods of CRC diagnosis, the authors may include Capsule endoscopy, CT colonography, and multitarget stool DNA test.  

Line 292, please change into CRC treatments instead of cancer treatments as this review is about colon cancer treatments instead of generalizing.

Under the treatments, please include discussions on adjuvant therapy and neoadjuvant therapy. 

Please include a paragraph on the future prospects of CRC treatment. Can gene therapy or gene editing technologies be effectively used? The authors can also include information on precision medicine, altering the gut microbiome, artificial intelligence, etc. 

Author Response

Comments and Suggestions for Authors

The authors in this review discuss on Colon cancer that is the third most frequent cancer and the second greatest cause of death worldwide. Despite advances in diagnosis and treatment, colorectal cancer (CRC) remains difficult to diagnose and treat, prompting this review to investigate its initiation, developmental stages, pathophysiology, risk factors, current diagnostics, treatment, and emerging advancements, emphasizing the importance of ongoing research and awareness on this critical issue. 

The authors have addressed all of the important limitations in CRC diagnosis and treatment options that are now available. Although there are now limitations to this, I feel that with the advancement of research, such restrictions can be addressed, particularly with regard to non-invasive means of diagnostic and treatment options. It would be interesting to see if artificial intelligence may be employed for CRC screening in the future. This review is sure to grab the readers' interest. Here are my thoughts.

  1. Figure 6: check the spelling of “surgery”

Response: The spelling is corrected

  1. 2 Chemotherapy: Mention the stage specific drugs such as the combination of 5FU, leucovorin, and oxaliplatin may be prescribed for stage III. 

Response: Thank you for the comment. We have given list of drugs for different stages for colorectal cancer treatment.

Bevacizumab (Avastin)- With fluorouracil as the first or second treatment.

With a fluoropyrimidine and either irinotecan hydrochloride or oxaliplatin as the second treatment in patients whose disease has gotten worse after therapy that included bevacizumab in Stage III patients.

Cetuximab (Erbitux)- In patients whose cancer has the EGFR protein and the wild-type KRAS gene. It is used with FOLFIRI combination chemotherapy as the first treatment.

With irinotecan hydrochloride in patients whose cancer was treated with chemotherapy that included irinotecan hydrochloride but it did not work or is no longer working. Alone in patients whose cancer did not respond to oxaliplatin and irinotecan hydrochloride or who cannot be treated with irinotecan hydrochloride.

Encorafenib (Braftovi)- Encorafenib is approved to be used with other drugs to treat patients whose cancer has a certain mutation in the BRAF gene. Colorectal cancer that has spread to other parts of the body. It is used with cetuximab in adults who have received previous treatment.

Ipilimumab (Yervoy): In adults and children aged 12 years and older. Ipilimumab is used with nivolumab to treat metastatic microsatellite instability-high (MSI-H) or mismatch repair deficient (dMMR) cancer that got worse after treatment with a fluoropyrimidine, oxaliplatin, and irinotecan hydrochloride.

National Cancer Institute Feb 24, 2023. https://www.cancer.gov/about-

  1. The use of phototherapeutic-based nanoparticles in colorectal cancer treatment is a unique and promising method, but like many medical therapies, may have negative effects. It is critical to recognize that the possible adverse effects might vary based on the type of nanoparticles employed, the patient's particular features, and the treatment procedures. Please include a discussion on this

Response: Perioperative treatment combined with radical resection is the major approach to cure non-metastatic colon cancer. A precise evaluation and perioperative treatment would probably improve the R0 resection rate, recurrence-free survival and overall survival of colon cancer patients. Recently, individualized treatment is the mainstream due to the development of molecular pathology and multi-disciplinary therapy. The indications and course of perioperative treatment and preoperative neoadjuvant therapy of colon cancer are still in intense discussion. The present review will mainly discuss three topics. Firstly, the various reaction of adjuvant therapy to stage II colon cancer is caused by patients' heterogeneity. Choosing stratified treatment for these patients according to clinical and molecular pathological features is the future. Secondly, we discuss the adjuvant chemotherapy course for stage III colon cancer according to the Chinese Society of Clinical Oncology (CSCO) guideline and the progress of this field. Lastly, we summarize the status and significance of colon cancer neoadjuvant therapy.

  1. Please comment on this issue.  The complexity of hormone regulation inside the human body is one of the key problems. Female hormones, such as estrogen and progesterone, have complex interactions that vary widely from person to person. Because of these differences, developing a one-size-fits-all nanotherapy method for CRC can be difficult.

Response: Thank you for your comment. Please find our response. It is thought that developing a one-size-fits-all nanotherapy method for CRC is difficult. This concern should be studied from different views, one of them is that each hormone has its own specific receptors which differ in male than females, like estrogens have three categories for oestrogen receptors: two nuclear hormone receptors (ERα, Erβ) and G-protein coupled oestrogen receptor (GPR30) [Parks et al, 2011]. [Louie and, Sevigny, 2017]. Also, theses nuclear hormone receptors are expressed to multiple tissues in different degrees, the key ER subtype in natural colonic epithelium is Erβ.  Liu et al. [2017] found that higher GPR30 expression in CRC tissue is associated with a better survival compared to lower GPR30 expression. But future studies are needed to investigate whether these receptors can be used for CRC progress [Chen J, Iverson, 2012]. Testosterone has its own receptors, normal colonic epithelial tissue can express two types of androgen receptors (AR-A and AR-B), but CRC can express AR-A aonly to bind with dihydrotestosterone [Roshan et al 2016]. A G-protein coupled receptor can only express in some colonic cancer cells in vitro but not in normal colon cells [Gu et al, 2011]. Thus, hormonal nanotherapy need more studies to know how to exert different effects according to different receptors could be bound with.

  1. Please comment on this issue. Hormonal therapy may have side effects and hazards that must be carefully considered, including the possibility of deleterious effects on other physiological systems. As a result, while the notion of using female hormones for CRC nano treatment is exciting, it requires more study and refining to solve these deficiencies and fully realize its therapeutic potential.

Response. Thank you for the comment. In 2021, Bouras and his team conducted a systematic review literatures and meta-analysis study to search the relationship between endogenous levels of sex hormones and CRC potential risk to do more investigation on what has been described by earlier preclinical studies. But Bouras et al revealed that there were not connections for endogenous gonads steroid hormones in either males or post-menopausal females with CRC risk. Additionally, they confirmed for essential heterogeneity noticed in women. The result of this meta-analysis study did not agree existence of connections between pre-diagnostic levels of estradiol, testosterone, and SHBG with occurrence risk of CRC in both human males and post-menopausal females.

Estradiol (E2) and progesterone (P4) mono and in combination therapy for CRC have been done in male and female in both vitro and in vivo models as experimental researches. Exogenous Estradiol treatment and/ or reactivation with its ERβ markedly suppressed cell proliferation and stimulated apoptosis. Similarly, the suppression of ERα receptors revealed same antitumorigenic effect. Thus, E2 has dual different functions in CRC based on its nuclear receptors. P4 are rare, it was reported that in vitro and in vivo therapy with normal and synthetic progesterone were also connected with promising antitumor effect. While the therapeutic combination of E2 with P4 displayed supportive anticancer effect in comparison with monotherapy of each of them in male or female cell in vitro and in animals. Made the female gonads steroid hormones a novel and promising therapeutic management for CRC (Mahbub 2022.)

Recent study reported that expression of sex steroid receptors in tumors could act as prediction markers, also hormonal therapy can act as an alternative strategic plan for treating CRC, while effects can be exerted according many factors sex, clinical phase, and tumor site (Refaat et al, 2023). The safety of hormonal nantherapy on CRC and other body system are still need more researches.

  1. Please specify the “change” in the line 199. Instead, more specific words can be used such as gain, LOH etc. Include discussion on chromosomal instability under chromosomal biomarkers for CRC.

Response: Thank you for the comment. Loss of heterozygosity (LOH) at chromosome 18q frequently occurs late during colon cancer development and is inversely associated with microsatellite instability (MSI). 18q LOH has been reported to predict shorter survival in patients with colorectal cancer, whereas MSI-high status has been associated with superior prognosis.

  1. The gut microbiota is extremely diverse, with individual differences. Because of this variety, as well as the existence of non-cancer-related microbiota, identifying particular microbial markers linked with CRC might be difficult. While it can identify some CRC-associated microbial signatures, can it reliably identify all instances, which might result in findings that are either falsely positive or negative?

Response: Thank you for the comment. Dysbiosis of the gut microbiome has been associated with the development of colorectal cancer (CRC). Gut microbiota is involved in the metabolic transformations of dietary components into oncometabolites and tumor-suppressive metabolites that in turn affect CRC development. The alpha-bug hypothesis suggested that oncogenic bacteria such as enterotoxigenic Bacteroides fragilis (ETBF) induce the development of CRC through direct interactions with colonic epithelial cells and alterations of microbiota composition at the colorectal site. Escherichia coli, E. faecalis, F. nucleatum, and Streptococcus gallolyticus showed higher abundance whereas Bifidobacterium, Clostridium, Faecalibacterium, and Roseburia showed reduced abundance in CRC patients (Chattopadhyay et al., 2021).

  1. Under methods of CRC diagnosis, the authors may include Capsule endoscopy, CT colonography, and multitarget stool DNA test. 

Response: Thank you for the comment, we have included these in the revised manuscript.

Capsule endoscopy can be used to evaluate the esophagus, stomach, small intestine, and colon. It is ingested just like any other capsule and travels through the esophagus into the stomach. It then passes through the pyloric sphincter into the duodenum, jejunum, and ileum.

  1. Nakajima F, Furumatsu Y, Yurugi T, Amari Y, Iida T, Fukui T, Kuramoto T. Investigation of small intestinal lesions in dialysis patients using capsule endoscopy. Hemodial Int. 2019 Jan;23(1):77-80

CT Colonography is used to screen for cancers and other conditions affecting the colon. This study looks for significant growths, such as polyps, within your rectum and colon. Polyps are growths on the colon's lining that sometimes grow into cancers.

  1. Obaro AE, Burling DN, Plumb AA. Colon cancer screening with CT colonography: logistics, cost-effectiveness, efficiency and progress. Br J Radiol. 2018 Oct;91(1090):20180307. doi: 10.1259/bjr.20180307. Epub 2018 Jul 5. PMID: 29927637; PMCID: PMC6350489.

Multitarget stool DNA test: The multi-target fecal immunochemical test (FIT) and stool DNA test not only detects mutations associated with colorectal cancer, it also incorporates the FIT test to detect blood. The positive side of the test: Is performed at home. Detects specific colorectal cancer-related mutations.

  1. Kleinschmidt TK, Clements A, Parker MA, Scarcliff SD. Retrospective Review of Multitarget Stool DNA as a Screening Test for Colorectal Cancer. Am Surg. 2023 Apr;89(4):603-606. doi: 10.1177/00031348211031844. Epub 2021 Jul 18. PMID: 34278822.
  2. Line 292, please change into CRC treatments instead of cancer treatments as this review is about colon cancer treatments instead of generalizing.

Response: Thank you for the comment. The correction has been made

  1. Under the treatments, please include discussions on adjuvant therapy and neoadjuvant therapy. 

Response: Thank you for the comment. In patients with stage III colon cancer (CC), adjuvant chemotherapy with the combination of oxapliplatin to a fluoropyrimidine (FOLFOX or CAPOX) is a standard of care. The duration of treatment can be reduced from 6 months to 3 months, depending on the regimen, for patients at low risk of recurrence, without loss of effectiveness and allowing a significant reduction in the risk of cumulative sensitive neuropathy [Taieb and Gallois, 2020]. Neoadjuvant systemic therapy has many potential advantages over up-front surgery, including tumor down-staging, early treatment of micro-metastatic disease, and providing an in vivo test of tumor biology. Due to these advantages, neoadjuvant therapy is becoming the standard of care for an increasing number of tumor types. Currently, colon cancer patients are still routinely treated with up-front surgery, and neoadjuvant systemic therapy is not yet standard. Limitations to widespread use of neoadjuvant therapy have included inaccurate radiological staging, concerns about tumor progression while undergoing preoperative treatment rendering a patient incurable, and a lack of randomized data demonstrating benefit. Nevertheless, there is abundant interest in neoadjuvant chemotherapy, and a number of trials are under way. Initial follow up of the first phase III trial of neoadjuvant chemotherapy for colon cancer demonstrated tumor down-staging and suggested an improvement in disease-free survival with neoadjuvant chemotherapy [Body et al., 2021]

- Taieb J, Gallois C. Adjuvant Chemotherapy for Stage III Colon Cancer. Cancers (Basel). 2020 Sep 19;12(9):2679. doi: 10.3390/cancers12092679. PMID: 32961795; PMCID: PMC7564362.

- Body A, Prenen H, Latham S, Lam M, Tipping-Smith S, Raghunath A, Segelov E. The Role of Neoadjuvant Chemotherapy in Locally Advanced Colon Cancer. Cancer Manag Res. 2021 Mar 17;13:2567-2579. doi: 10.2147/CMAR.S262870. PMID: 33762848; PMCID: PMC7982559.

  1. Please include a paragraph on the future prospects of CRC treatment. Can gene therapy or gene editing technologies be effectively used? The authors can also include information on precision medicine, altering the gut microbiome, artificial intelligence, etc. 

Response: heparin-polyethyleneimine (HPEI) nanoparticles were used to deliver plasmid-expressing mouse survivin-T34A (ms-T34A) to treat colon cancer C-26 carcinoma in vitro and in vivo. According to the in vitro studies, HPEI nanoparticles could efficiently transfect the pGFP report gene into C-26 cells, with a transfection efficiency of 30.5% ± 2%.  Intratumoral injection of HPEI nanoparticle-mediated ms-T34A significantly inhibited growth of subcutaneous C-26 carcinoma in vivo by induction of apoptosis and inhibition of angiogenesis (Zhang et al., 2011). In another, it was found that that the gef gene induced a marked decrease in cell viability (50% in 24h) in T-84 cells through cell death by apoptosis. Interestingly, when gef gene expression was combined with drugs of choice in the clinical treatment of colon cancer (5-fluorouracil, oxaliplatin and irinotecan), a strong synergistic effect was observed with approximately a 15-20% enhancement of the anti-proliferative effect (Ortiz et al., 2012). In addition, reactivation of survivin expression is involved in carcinogenesis and angiogenesis in colon cancer. The therapeutic effect of the adeno-associated virus (AAV)-mediated survivin mutant (Cys84Ala) on colon cancer. It was reported that transduction of colon cancer cells with rAAV-Sur-Mut(Cys84Ala) inhibited cell proliferation and induced apoptosis and mitotic catastrophe in vitro. rAAV-Sur-Mut(Cys84Ala) sensitized colon cancer cells to chemotherapeutic drugs. Intratumoral injection of rAAV-Sur-Mut(Cys84Ala) significantly induced apoptosis and mitotic catastrophe and inhibited angiogenesis and tumor growth in a colon cancer xenograft model in vivo (Tu et al., 2005). Recently, the role of miR-34a and miR-34b/c encoding genes were in the colon cancer therapy (Huang et al., 2023). Dysbiosis of the gut microbiome has been associated with the development of colorectal cancer (CRC). Gut microbiota is involved in the metabolic transformations of dietary components into oncometabolites and tumor-suppressive metabolites that in turn affect CRC development. The alpha-bug hypothesis suggested that oncogenic bacteria such as enterotoxigenic Bacteroides fragilis (ETBF) induce the development of CRC through direct interactions with colonic epithelial cells and alterations of microbiota composition at the colorectal site. Escherichia coli, E. faecalis, F. nucleatum, and Streptococcus gallolyticus showed higher abundance whereas Bifidobacterium, Clostridium, Faecalibacterium, and Roseburia showed reduced abundance in CRC patients (Chattopadhyay et al., 2021). The application of exogenous Lactobacillus reuteri restricts colon tumor growth, increases tumor reactive oxygen species, and decreases protein translation in vivo (Bell et al., 2022).

  1. Zhang L, Gao X, Men K, Wang B, Zhang S, Qiu J, Huang M, Gou M, Huang N, Qian Z, Zhao X, Wei Y. Gene therapy for C-26 colon cancer using heparin-polyethyleneimine nanoparticle-mediated survivin T34A. Int J Nanomedicine. 2011;6:2419-27. doi: 10.2147/IJN.S23582. Epub 2011 Oct 19. PMID: 22072877; PMCID: PMC3205136.
  2. Ortiz R, Prados J, Melguizo C, Rama AR, Alvarez PJ, Rodríguez-Serrano F, Caba O, Boulaiz H, Aranega A. Gef gene therapy enhances the therapeutic efficacy of cytotoxics in colon cancer cells. Biomed Pharmacother. 2012 Oct;66(7):563-7. doi: 10.1016/j.biopha.2012.05.004. Epub 2012 Jun 19. PMID: 22770988.
  3. Tu SP, Cui JT, Liston P, Huajiang X, Xu R, Lin MC, Zhu YB, Zou B, Ng SS, Jiang SH, Xia HH, Wong WM, Chan AO, Yuen MF, Lam SK, Kung HF, Wong BC. Gene therapy for colon cancer by adeno-associated viral vector-mediated transfer of survivin Cys84Ala mutant. Gastroenterology. 2005 Feb;128(2):361-75. doi: 10.1053/j.gastro.2004.11.058. PMID: 15685548.
  4. Huang Z, Kaller M, Hermeking H. CRISPR/Cas9-mediated inactivation of miR-34a and miR-34b/c in HCT116 colorectal cancer cells: comprehensive characterization after exposure to 5-FU reveals EMT and autophagy as key processes regulated by miR-34. Cell Death Differ. 2023 Aug;30(8):2017-2034. doi: 10.1038/s41418-023-01193-2. Epub 2023 Jul 24. PMID: 37488217; PMCID: PMC10406948.
  5. Chattopadhyay I, Dhar R, Pethusamy K, Seethy A, Srivastava T, Sah R, Sharma J, Karmakar S. Exploring the Role of Gut Microbiome in Colon Cancer. Appl Biochem Biotechnol. 2021 Jun;193(6):1780-1799. doi: 10.1007/s12010-021-03498-9. Epub 2021 Jan 25. PMID: 33492552.
  6. Bell HN, Rebernick RJ, Goyert J, Singhal R, Kuljanin M, Kerk SA, Huang W, Das NK, Andren A, Solanki S, Miller SL, Todd PK, Fearon ER, Lyssiotis CA, Gygi SP, Mancias JD, Shah YM. Reuterin in the healthy gut microbiome suppresses colorectal cancer growth through altering redox balance. Cancer Cell. 2022 Feb 14;40(2):185-200.e6. doi: 10.1016/j.ccell.2021.12.001. Epub 2021 Dec 23. PMID: 34951957; PMCID: PMC8847337.

Round 2

Reviewer 1 Report

the authors responded point by point to the comments and questions, which significantly improved the quality of the revised manuscript.